# Future Healthcare Workers and Hepatitis B Vaccination: A New Generation

**DOI:** 10.3390/ijerph18157783

**Published:** 2021-07-22

**Authors:** Andrea Trevisan, Paola Mason, Annamaria Nicolli, Stefano Maso, Marco Fonzo, Bruno Scarpa, Chiara Bertoncello

**Affiliations:** 1Department of Cardiac Thoracic Vascular Sciences and Public Health, University of Padova, Via Giustiniani 2, 35128 Padova, Italy; paola.mason.1@unipd.it (P.M.); annamaria.nicolli@unipd.it (A.N.); stefano.maso@unipd.it (S.M.); marco.fonzo@unipd.it (M.F.); chiara.bertoncello@unipd.it (C.B.); 2Department of Statistical Sciences, University of Padova, 35128 Padova, Italy; bruno.scarpa@unipd.it

**Keywords:** hepatitis B vaccination, healthcare workers, students, antibody titer, age at vaccination

## Abstract

Before the introduction of universal vaccination, hepatitis B caused high morbidity and mortality, especially among healthcare workers. In the present study, the immune status against hepatitis B was assessed in a cohort of 11,188 students of the degree courses of the School of Medicine of the University of Padua (Italy) who had been subjected to mandatory vaccination in childhood or adolescence and who will be future healthcare workers. The variables that influence the antibody response to vaccination are mainly the age at which the vaccine was administered and sex. If vaccination was administered before one year of age, there is a high probability (around 50%) of having an antibody titer lower than 10 IU/L compared to those vaccinated after one year of age (12.8%). The time between vaccine and analysis is not decisive. Furthermore, female sex, but only if vaccination was administered after one year of age, shows a significant (*p* = 0.0008) lower percentage of anti-HBs below 10 IU/L and a greater antibody titer (*p* < 0.0001). In conclusion, the differences related to the age of vaccination induce more doubts than answers. The only plausible hypothesis, in addition to the different immune responses (innate and adaptive), is the type of vaccine. This is not easy to verify because vaccination certificates rarely report it.

## 1. Introduction

Hepatitis B (HBV) is a blood-borne transmittable infectious disease that before the introduction of universal vaccination (initially a plasma-derived and a successively genetically engineered *Saccharomyces cerevisiae* yeast recombinant vaccine) caused high levels of morbidity and mortality, especially among healthcare workers (HCWs) [1]. On the other hand, the attitude of HCWs to vaccination appears suboptimal [2,3].

With the implementation of mandatory vaccination in Italy in 1991 (for all newborns and adolescents of 12 years of age), morbidity has drastically reduced, especially in the more vulnerable age groups [4], passing from a country with medium endemicity to one with low endemicity.

The new generation of HCWs, those who have been mandatorily vaccinated since 1980, are now covered by the vaccine and, for the most part, they are also immune.

As it has been pointed out, a mandatory HBV vaccination campaign appears to be the best method for inducing immunity [5], but in Europe, mandatory vaccination is not homogeneous for newborns nor for HCWs [6].

Numerous research studies highlight the fact that mass vaccination induces a prolonged protective action [7,8,9,10,11,12] and how the presence of antibodies lower than 10 IU/L, a value considered the cut-off between protection and non-protection, is not necessarily the cause of infection [9,13,14]. The attitude for vaccination against HBV and the anti-HBs titer in HCWs and in students has been evaluated in recent years and the majority of subjects investigated were protected even after many years [15,16,17,18,19,20,21].

Among the unprotected subjects, it is necessary to distinguish the so-called non-responders who remain, fortunately limited to a modest percentage of cases, a problem; they are identified as someone who, as indicated by the Center for Disease Control and Prevention [22], “does not develop protective surface antibodies after completing two full series of the hepatitis B vaccine and for whom an acute or chronic hepatitis B infection has not been ruled out”. The aim of this study was to assess the antibody titer against hepatitis B in the future generation of HCWs at the time of their university enrollment and to evaluate the effect of age at vaccination, sex, year of birth, and time between vaccination and antibody analysis on the antibody titer itself.

## 2. Materials and Methods

### 2.1. Population

The study enrolled 11,188 students (2004–2020) attending the Medical School (medicine and surgery, dentistry, and healthcare professions) of Padua University (Northern Italy). The enrollment criteria were that they had to be born in Italy and then vaccinated with the same schedule (three doses) and to present a certificate from the Public Health Office certifying the vaccination against HBV. The students attended graduate courses in medicine and surgery (5121, 45.8%), dentistry (337, 3.0%), and healthcare professions (5730, 51.2%). There were almost twice the number of females (7291, 65.2%) than males (3897, 34.8%), with the rate of males/females being 0.53. The preponderance of the female sex is mainly due to the high presence of women in the study courses of healthcare professions (ratio 0.34) rather than in those of medicine and surgery (ratio 0.79) or dentistry (ratio 1.22).

The students were mainly from Northern Italy (93.2%), in the Veneto Region (85.0%), and, in particular, the Province of Padua (32.3%). Those from Central and Southern Italy (1.8% and 5.0%, respectively) mainly attended the degree course in medicine. Figure 1 represents Italy and its subdivision into Northern, Central, and Southern regions.

The students were further subdivided according to the age of vaccination (before one year of age or after) and into four-year birth groups (born between 1980 and 1985, between 1986 and 1990, between 1991 and 1995, and after 1995) to better highlight the differences in response to the vaccine according to the age of vaccination, which depends on the year of birth.

### 2.2. Antibody Measurement

A commercial chemiluminescent microparticle immunoassay (CMIA) was used until 2017 to measure the anti-HBs titer. After that, at the end of 2017, the clinical microbiology laboratory changed the instrumentations and adopted a different commercial kit. The procedure uses a chemiluminescent immunoassay (CLIA) named LIAISON^®^ anti-HBs plus by Sorin (Saluggia, Italy) and the lower cut-off is expressed only as below 3 IU/L without the possibility to know the values hidden under this value. For this reason, values indicated as below 3 IU/L were processed as 3 IU/L. Similarly, the method does not provide values higher than 1000 IU/L and, therefore, for a statistical evaluation of the antibody titer, they were processed as 1000 IU/L.

### 2.3. Statistics

A chi-squared (χ^2^) test 2 by 2 (Yates’ correction) was used to compare the frequencies of anti-HBs titers below 10 IU/L between different vaccination strategies and between year of birth. The presence of antibodies lower than 10 IU/L is considered the cut-off between protection and non-protection. Multiple linear regression based on the logarithm of the antibody titer (being asymmetric in its distribution) was employed to analyze the variables influencing antibody level (dependent variable). The following outcomes were considered for the purpose of multiple linear regression if appropriate (independent variables): (1) sex, (2) year of birth, and (3) time between the last dose of vaccine and analysis of antibodies. Statistical differences among antibody titers according to the age of vaccination and sex were evaluated by means of an unpaired t-test assuming unequal variance. In order to better highlight the possible differences between those vaccinated before and after one year of age, the subjects whose time between vaccine and analysis was between 16 and 25 years were extrapolated from both cohorts. The differences in terms of antibody titer and the prevalence of antibody levels lower than 10 IU/L were assessed with a parametric test as above for the former and with a chi-squared (χ^2^) test for the latter. Other statistical analyses were descriptive. Significance is stated by *p* < 0.05. Statsdirect 2.7.7 version (Statsdirect Ltd., Birkenhead, Merseyside, UK) was used for the statistical analyses.

## 3. Results

The influence of sex and age at vaccination on the antibody response is shown in Table 1, where a significant difference is observed in all students both as a percentage of anti-HBs lower than 10 IU/L (*p* = 0.0008) and the antibody titer (*p* < 0.0001) but only if they were vaccinated after the first year of age (*p* < 0.0001 for both parameters).

Figure 2 illustrates how the antibody titer and the prevalence of subjects below 10 IU/L is related to the year of birth (1991–1992) in which infants were vaccinated at three months of age.

To define once and for all whether the interval between the vaccine and analysis is determined to cause both a consistent reduction in the antibody titer and a high percentage of subjects with a titer lower than 10 IU/L, f those with the same time between 16 and 25 years were extrapolated from the context (Table 2). With the same time, the statistically significant difference (*p* < 0.0001) regards the percentage of antibodies lower than 10 IU/L and the antibody titer remains in favor of those vaccinated after one year of age.

Further, between those vaccinated before and after one year of age, there are individuals (relatively few) who are outside the canonical ages established by the law. In both cases, birth before 1990 shows, although not always significantly, a lower percentage of anti-HBs below 10 IU/L and a higher antibody titer (Table 3).

Multiple linear regression based on the logarithm of the antibody titer identified three independent variables, namely year of birth, sex, and time between the third dose of the vaccine and analysis of antibodies.

Overall, year of birth and time between vaccine and analysis significantly affect (*p* < 0.0001) the antibody response, whereas sex is only significant if students were vaccinated after the first year of age (Table 4).

## 4. Discussion

In 1992, the WHO included HBV vaccination to among those recommended, but in Italy, the mandatory mass vaccination had already started in 1991 for newborns and adolescents. Incomprehensibly, no vaccinations are mandatory in Italy for HCWs, not even that against HBV. Fortunately, given the introduction of mandatory vaccination for those born since 1980, we have a new generation and will have future generations of vaccinated HCWs.

Vaccination against HBV is safe, and immunity lasts a long time [7,12,21,23]. A cut-off of 10 IU/L is the watershed to consider antibody levels protective [14], although it has been widely demonstrated that immune memory persists despite very low antibody titers [24,25,26].

The results of this study highlight that there are many variables that can influence the level of antibodies against HBV. The year of birth and consequently the age of administration of the first dose of the vaccine is the variable most influencing the antibody response.

We have long argued that vaccination administered before one year of age has, as a consequence, a high percentage of subjects with antibody levels lower than 10 IU/L and a low antibody titer [27]. The current results, on a 10-times higher number of cases, confirm this. The obvious objection is that the time between the vaccine and antibody analysis is almost double that between the two cohorts. To dispel any doubts, the two cohorts were again compared in relation to the time and the results were comparable to the previous ones [27].

If time is not the deciding factor, why are there such discrepancies? We tried to test (unfortunately the numbers are small) whether there was a difference between the two cohorts in relation to the year of birth. The difference is evident above all for the antibody titer. Vaccination after one year of age induces a higher antibody titer (independently by the time elapsed between the last vaccine dose and the antibody analysis), probably in relation to the development of adaptive immunity.

Another hypothesis, which could be supported by the evidence that birth in the decade 1980–1990, even if vaccinated before one year, gives better results, in particular for the antibody titer, is the type of vaccine. As already highlighted [25,28], the Engerix-B^®^ vaccine is far more immunogenic than the Recombivax HB^®^ vaccine, even in the hexavalent formulation; however, the latter came into use in the mid-eighties and was subsequently replaced by others [29]. The lower efficacy of the Recombivax HB^®^ vaccine has been well documented [29,30]. Currently, there are no other acceptable explanations, and this is surely a weakness of the study.

Despite this, the disappearance of the circulating antibodies does not signify a loss of the immune response [31]. The activation of T lymphocytes permits HBsAg-specific memory persistence [32]. Protection induced by the HBV vaccine has been associated with specific SNPs encoding cytokines and TLR2. On the contrary, non-responders more frequently express SNPs rs1 143633 (IL-1B; intron) and rs1 143627 (IL-1B; intron) that define the AG haplotype [33].

In detail, the polymorphism of the IL4 gene is fundamental for the response to the vaccine [34], while polymorphisms of the immunoregulatory cytokines and *HLA-DRB1*07* genes cause a variable response to recombinant vaccines [35]. The scenario in the case of non-response to the HBV vaccine appears more complex. The causes can be ascribed to adequate production of Th1-and Th2-like cytokines [36], changes in the expression in CXCR5 associated with polymorphisms in follicular helper T cells [37], and the TT genotype of the *IFNG* (+874 T/A) gene and alleles 170 and 182 for the (CA)n alleles for the intronic (CA)n microsatellite of the *IFNGR1* gene [38]. A prompt response after a booster dose reveals an effective immune memory [8,39], whereas low antibody production suggests a loss of immune memory [40]. Most subjects with a non-protective antibody titer (<10 IU/L) produce anti-HBs after the booster due to activation of B cells [41]. In contrast, non-responders have a low B cell response [42]. Further, immune memory persists for at least five years in children vaccinated as infants [43].

Finally, the response to a booster dose is similar in the two cohorts [25,26]. Thus, maintaining long-term protection from a routine booster is unnecessary [44].

Recently [45], we suggested a different response to the HBV vaccine related to sex, and also after a booster dose [26]. However, a significant difference was observed only in those vaccinated after one year of age in terms of the antibody titer and values below 10 IU/L. In fact, although innate immunity is similar [46], adaptive immunity is more pronounced in females, and this justifies the discrepancy between the two cohorts. Adaptive immunity depends on three fundamental components, such as B cells, CD4^+^ T cells, and CD8^+^ T cells [47]. As estrogens have a high role in adaptive immunity [48], a major response in females appears an obvious consequence.

Finally, measurement of the anti-HBs titer, at least during the first health surveillance, is useful to identify the (few) HCWs with antibody titer below that considered protective [15,16,17].

This study has some limitations. First, we were not able to retrieve the data on the type of vaccine administered (either Engerix-B^®^ or Recombivax HB^®^) at the individual level, and second, the method used to assess the antibody titers does not provide values higher than 1000 IU/L and this may have affected statistical analyses.

## 5. Conclusions

The results of this research pose many questions and the answers are neither easy nor definitive. The main question that has tormented us for years is why vaccinating before one year of age, even with the same time between the vaccine and analysis, shows a remarkable difference both in terms of the antibody titer and the prevalence of antibodies lower than 10 IU/L. The type of vaccine could be responsible for the observed differences, but we do not know for certain because the vaccine certificates do not report the type. The lack of knowledge of the type of vaccine is certainly a weakness of the study. However, we suggest that in low endemicity countries, HBV vaccination should be performed after one year of age.

## Figures and Tables

**Figure 1 ijerph-18-07783-f001:**
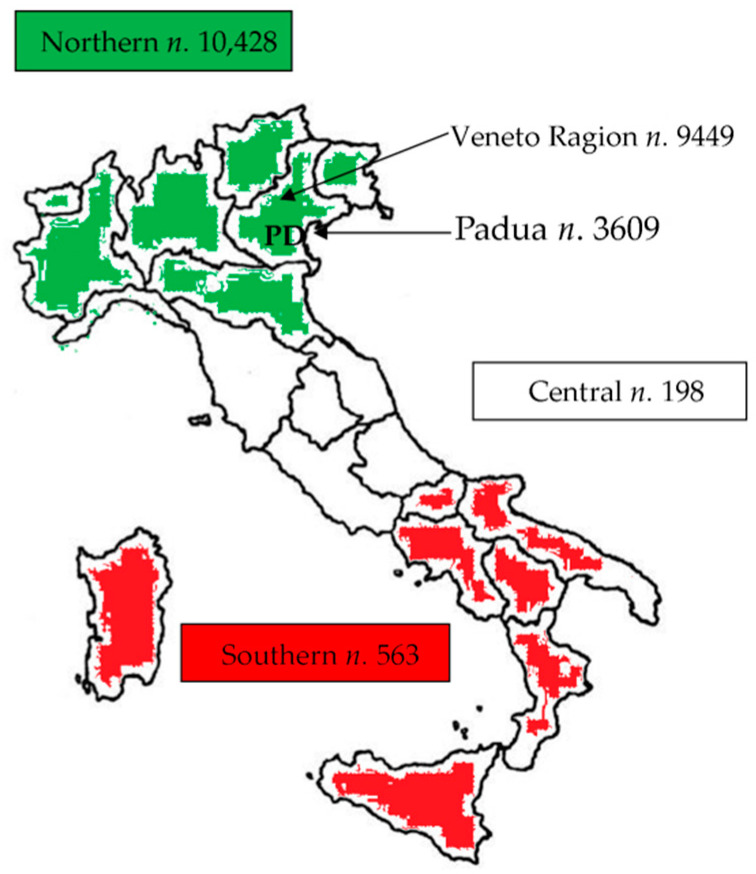
Representation of Italy divided into Northern (green), Central (white), and Southern (red) regions. The numbers of students from each of the three areas and in particular from the Veneto Region and the Province of Padua are specifically highlighted.

**Figure 2 ijerph-18-07783-f002:**
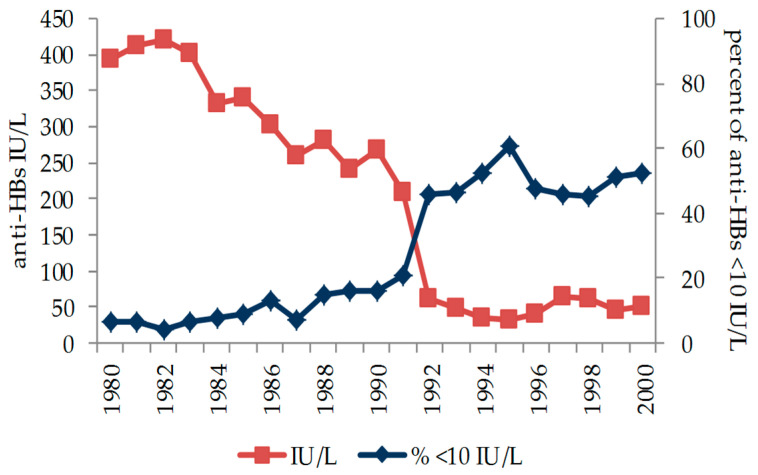
Antibody titer level (IU/L) and antibody titer lower than 10 IU/L according to year of birth 1980–2000. Those born before 1 January 1992 were vaccinated at 12 years of age, with those born after being vaccinated at 3 months of age, in compliance with the vaccination schedule adopted at the beginning of the mandatory vaccination campaign against HBV.

**Table 1 ijerph-18-07783-t001:** Seroprevalence of anti-HBs lower than 10 IU/L and antibody titer for all students and the two cohorts vaccinated before or after the first year of age according to sex also. The last column shows the time between the last vaccine dose and the antibody analysis.

Students	*n*	<10 IU/L	%	χ^2^	*p*	Titer IU/L Mean ± SD	*p*	Time Mean ± SD
All students	11,188	3679	32.9	11.144	=0.0008 *	166.0 ± 288.7	<0.0001 *	15.3 ± 4.9
males	3897	1361	34.9			143.3 ± 265.5		15.9 ± 4.7
females	7291	2318	31.8			178.2 ± 299.8		15.1 ± 5.0
Age at vaccination								
Vaccinated before 1 year of age	6013	3017	50.2	1759.371	<0.0001 **	50.1 ± 127.9	<0.0001 **	19.4 ± 1.1
males	2179	1080	49.6	0.472	n.s.	51.8 ± 129.7	n.s.	19.5 ± 1.0
females	3834	1937	50.5			49.1 ± 126.8		19.3 ± 1.1
Vaccinated after 1 year of age	5175	662	12.8			300.7 ± 357.1		10.7 ± 3.2
males	1718	281	16.4	28.805	<0.0001 *	259.3 ± 338.4	<0.0001 *	11.3 ± 3.2
females	3457	381	11.0			321.3 ± 364.3		10.3 ± 3.1

* Statistical significance of the differences between males and females, where females have a lower percentage of anti-HBs lower than 10 IU/L and a higher antibody titer; ** statistical significance of the differences between all students vaccinated before or after one year of age. n.s. = not significant

**Table 2 ijerph-18-07783-t002:** Seroprevalence of anti-HBs lower than 10 IU/L and titer according to vaccination before and after the first year of age with a time between vaccine and analysis of antibodies between 16 and 25 years.

Time All ≥ 16 Years	*n*	<10 IU/L	%	χ^2^	*p*	Titer IU/L Mean ± SD	*p*
vaccinated before 1 year	6013	3017	50.2	165.235	<0.0001	50.1 ± 127.9	<0.0001
vaccinated after 1 year	383	62	16.2			222.0 ± 314.7	

**Table 3 ijerph-18-07783-t003:** Seroprevalence of anti-HBs lower than 10 IU/L and titer according to vaccination before and after the first year of age and date of birth between 1980 and 1990 or after.

Vaccination	*n*	<10 IU/L	%	χ^2^	*p*	Titer IU/L Mean ± SD	*p*	Time Mean ± SD
before 1 year of age				0.273	n.s.			
born 1980–1990	36	16	44.4			114.9 ± 254.9		20.1 ± 1.9
born in 1991 or after	5977	3001	50.2			49.7 ± 126.7	=0.0023	19.4 ± 1.0
after 1 year of age				11.825	=0.0006			
born 1980–1990	4573	558	12.2			308.6 ± 360.3		10.7 ± 3.1
born in 1991 or after	602	104	17.3			241.1 ± 325.7	<0.0001	10.1 ± 3.5

**Table 4 ijerph-18-07783-t004:** Multiple linear regression on the influence of independent variables on the anti-HBs titer (logarithmic transformation) in all students and in two cohorts of those vaccinated before or after the first year of age. Significant results are in bold.

**All Students**	**b**	***r***	**t**	***p***
Intercept	101.326457		23.188326	<0.0001
year of birth	−0.049739	−0.208528	−22.548445	<0.0001
sex	0.031555	0.017194	1.818617	=0.069
time	−0.000152	−0.228768	−24.852321	<0.0001
**Vaccinated before 1 year of age**	**b**	***r***	**t**	***p***
Intercept	−34.537893		−3.868822	<0.0001
year of birth	0.017517	0.050819	3.944496	<0.0001
sex	−0.035231	−0.019884	−1.541681	=0.1232
time	0.000086	0.036904	2.862629	=0.0042
**Vaccinated after 1 year of age**	**b**	***r***	**t**	***p***
Intercept	110.800937		13.20865	<0.0001
year of birth	−0.054679	−0.17756	−12.974467	<0.0001
sex	0.148749	0.079279	5.718946	<0.0001
time	−0.000075	−0.092701	−6.694957	<0.0001

## Data Availability

Raw data are available upon request from the corresponding author.

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
