# Peer review of "Future Healthcare Workers and Hepatitis B Vaccination: A New Generation"

_ijerph, 2021, doi:10.3390/ijerph18157783_

Round 1
Reviewer 1 Report
The manuscript entitled Future Healthcare Workers and Hepatitis B Vaccination: A New Generation evaluates hepatitis B virus (HBV) immunization status and anti-HBs titer among future healthcare workers (HCWs) who are at high risk for HBV infection.
The unquestionable strength of the study is that the topic is interesting and the research itself was designed and described clearly. Moreover, as it identifies the main variables that affect the levels of antibodies among those vaccinated subjects: year of birth, sex, and time between third dose of vaccine and analysis of antibodies, it provides insights for the future vaccination programmes and has great value in determining the most effective time for HBV vaccination process in HCW. Finally, the study was conducted on a large number of participants.
At the same time, my biggest concerns relates to the originality/novelty of the study as there are multiple research on the topic, some of which were also conducted in Italy, i.e. Verso et al. 2017; Coppeta et al. 2019; Cocchio et al. 2021.
Apart from that I have only some minor comments that could be elaborated more deeply before it could be considered for publication:
1. In the ‘Introduction’ some more information regarding knowledge and attitudes of HW and medical students towards HBV vaccination could be discussed. Moreover, I would suggest adding some more details on the effects of the vaccination in the population under study.
2. Although the paper cites many valuable research some additional reading could be added, i.e.:
-- Batra V, Goswami A, Dadhich S, Kothari D, Bhargava N. Hepatitis B immunization in healthcare workers. Ann Gastroenterol. 2015;28(2):276-280.
-- Lamberti M, De Rosa A, Garzillo EM, Corvino AR, Sannolo N, De Pascalis S, Di Fiore E, Westermann C, Arnese A, Gabriella DG, Nienhaus A, Sobrinho APR, Coppola N. Vaccination against hepatitis b virus: are Italian medical students sufficiently protected after the public vaccination programme? J Occup Med Toxicol. 2015;10:41. doi: 10.1186/s12995-015-0083-4.
-- Verso MG, Lo Cascio N, Noto Laddeca E, Amodio E, Currieri M, Giammanco G, Ferraro D, De Grazia S, Picciotto D. Predictors of Hepatitis B Surface Antigen Titers two decades after vaccination in a cohort of students and post-graduates of the Medical School at the University of Palermo, Italy. Ann Agric Environ Med. 2017;24(2):303-306. doi: 10.26444/aaem/74716.
-- Coppeta L, Pompei A, Balbi O, Zordo LM, Mormone F, Policardo S, Lieto P, Pietroiusti A, Magrini A. Persistence of Immunity for Hepatitis B Virus among Heathcare Workers and Italian Medical Students 20 Years after Vaccination. Int J Environ Res Public Health. 2019;16(9):1515. doi: 10.3390/ijerph16091515.
-- Hiva, S., Negar, K., Mohammad-Reza, P. et al. High level of vaccination and protection against hepatitis B with low rate of HCV infection markers among hospital health care personnel in north of Iran: a cross-sectional study. BMC Public Health 20, 920 (2020). doi: 10.1186/s12889-020-09032-6.
-- van Leeuwen LPM, Doornekamp L, Goeijenbier S, de Jong W, de Jager HJ, van Gorp ECM, Goeijenbier M. Evaluation of the Hepatitis B Vaccination Programme in Medical Students in a Dutch University Hospital. Vaccines. 2021;9(2):69. doi: 10.3390/vaccines9020069.
-- Cocchio S, Baldo V, Volpin A, Fonzo M, Floreani A, Furlan P, Mason P, Trevisan A, Scapellato ML. Persistence of Anti-Hbs after up to 30 Years in Health Care Workers Vaccinated against Hepatitis B Virus. Vaccines. 2021;9(4):323. doi: 10.3390/vaccines9040323.
All in all, although I am concerned over the novelty/originality of the study I found the paper interesting and timely and I believe that after some minor revision it could be published.
Author Response
Reviewer 1
We are grateful to the reviewer for his suggestions which are sure to make improvements to the manuscript.
The manuscript entitled Future Healthcare Workers and Hepatitis B Vaccination: A New Generation evaluates hepatitis B virus (HBV) immunization status and anti-HBs titer among future healthcare workers (HCWs) who are at high risk for HBV infection.
The unquestionable strength of the study is that the topic is interesting and the research itself was designed and described clearly. Moreover, as it identifies the main variables that affect the levels of antibodies among those vaccinated subjects: year of birth, sex, and time between third dose of vaccine and analysis of antibodies, it provides insights for the future vaccination programmes and has great value in determining the most effective time for HBV vaccination process in HCW. Finally, the study was conducted on a large number of participants. Apart from that I have only some minor comments that could be elaborated more deeply before it could be considered for publication:
Reply: We are grateful to the reviewer for his comments. Certainly writing about hepatitis B serology is not new, but we are sure it is necessary that we arrive at a shared vision of the problem. HCWs must necessarily undergo surveillance (I am a proud believer in the need to make certain vaccinations mandatory for HCWs).
- In the ‘Introduction’ some more information regarding knowledge and attitudes of HW and medical students towards HBV vaccination could be discussed. Moreover, I would suggest adding some more details on the effects of the vaccination in the population under study.
Reply: The recommended references have been included.
All in all, although I am concerned over the novelty/originality of the study I found the paper interesting and timely and I believe that after some minor revision it could be published.
Reply: We are grateful to the reviewer for her/his comments.
Reviewer 2 Report
Authors evaluated people vaccinated with the Hepatitis B vaccine in Italy and determined that sex and age at vaccination influence antibody response in individuals. However, the text is hard to read and contains conflicting statements. Authors should revise the text.
- There are several grammatical errors throughout the text, and these errors make the text very difficult to read and understand. Authors should have a native English speaker proofread their manuscript.
- There are conflicting statements in the text. For example, authors should fix this confusing sentence (shouldn’t the group with the higher antibody titer be the most immunogenic?). “Subjects born between 1980 and 1990 have a higher antibody titre (2.3 times if vaccinated before one year of age but 1.3 if vaccinated after). The vaccination after one year of age is therefore more immunogenic, probably in relation to the development of adaptive immunity”
Author Response
Reviewer 2
We are grateful to the reviewer for his suggestions which are sure to make improvements to the manuscript.
Comments and Suggestions for Authors
Authors evaluated people vaccinated with the Hepatitis B vaccine in Italy and determined that sex and age at vaccination influence antibody response in individuals. However, the text is hard to read and contains conflicting statements. Authors should revise the text.
- There are several grammatical errors throughout the text, and these errors make the text very difficult to read and understand. Authors should have a native English speaker proofread their manuscript.
Reply: The manuscript has been submitted for English editing and the new version is consistent with the corrections suggested by the language reviewer.
- There are conflicting statements in the text. For example, authors should fix this confusing sentence (shouldn’t the group with the higher antibody titer be the most immunogenic?). “Subjects born between 1980 and 1990 have a higher antibody titre (2.3 times if vaccinated before one year of age but 1.3 if vaccinated after). The vaccination after one year of age is therefore more immunogenic, probably in relation to the development of adaptive immunity”
Reply: All the conflicting statements have been clarified. The sentence has been replaced with this: The vaccination after one year of age induces a higher antibody titer (independently by the time elapsed between the last vaccine dose and the antibody analysis), probably in relation to the development of adaptive immunity.
Round 2
Reviewer 2 Report
Authors addressed reviewer's concerns. The manuscript is now acceptable for publication.